# Validation of the Vietnamese version of the Montgomery-Asberg depression rating scale (MADRS) in rheumatoid arthritis patients

Ha Thi Thu Tran[1,2]*, Long Bao Hoang[3], Hung Van Nguyen[4,5], Tuan Van Nguyen[1,2], Ha Thi Thu Le[1,2], Yen Hoang Nguyen[1,2], Hoa Thi Nguyen[1,2], Hue Thi Doan[1,2], Khiem Tuan Ngo[2], Thien Cong Le[1,2], Cam Thy Vu[2], Duong Thuy Lai[4], Tung Son Vu[1,2], Long Thanh Nguyen[1,2], Nga Thi Nguyet Pham[1], Tam Minh Duong[1,2]

1 Department of Psychiatry, Hanoi Medical University, Hanoi, Vietnam, 2 Vietnam National Institute of Mental Health, Hanoi, Vietnam, 3 College of Health Sciences, VinUniversity, Hanoi, Vietnam, 4 Rheumatology Center, Bach Mai Hospital, Hanoi, Vietnam, 5 Internal Medicine Department, Hanoi Medical University, Hanoi, Vietnam

* tran_thuha@hmu.edu.vn

## Abstract

This study aimed to validate the Vietnamese version of the Montgomery-Asberg Depression Rating Scale (MADRS) in patients with rheumatoid arthritis (RA). A cross-sectional study was conducted on RA patients, with depression severity assessed using both MADRS and the Patient Health Questionnaire-9 (PHQ-9). Internal consistency was evaluated using Cronbach's alpha, and the correlation between MADRS and PHQ-9 was measured using Spearman's correlation. Face validity was established through a back-translation process to ensure language equivalence. The translated version underwent exploratory and confirmatory factor analysis to evaluate its construct validity. The Vietnamese MADRS demonstrated excellent internal consistency (Cronbach's alpha = 0.93) and a strong linear correlation with PHQ-9 (Spearman's r = 0.88). Exploratory factor analysis identified a two-factor structure, categorizing depressive symptoms into general and severe clusters, which was confirmed in confirmatory factor analysis with borderline goodness of fit. Despite limitations such as sample specificity and the absence of a gold standard diagnostic tool, the study suggests that the Vietnamese MADRS can be effectively used in clinical and research settings in Vietnam for assessing depression in RA patients.

## Introduction

Rheumatoid arthritis (RA) is a chronic autoimmune disease that affects 0.5–1% of the population, with significant consequences for both physical and mental health [1]. Patients with RA experience depression at a prevalence of two to four times higher than the general population, with depressive symptoms often closely associated with pain and disease activity [2–5]. The presence of depression in RA not only worsens RA symptoms but also

which permits unrestricted use, distribution, and reproduction in any medium, provided the original author and source are credited.

**Data availability statement:** Data is freely accessible via the following link: https://doi.org/10.7910/DVN/YOLQTS.

**Funding:** The authors received no specific funding for this work.

has serious implications for quality of life, treatment adherence, and even an increased risk of suicide [6–8]. These factors highlight the urgent need for early screening and effective management of depression in RA population, with psychological instruments playing a key role in evaluating the severity of depressive symptoms. The Montgomery-Asberg Depression Rating Scale (MADRS) is one such tool, widely used for its brief yet effective assessment of depressive symptoms across different severity levels [9].

To ensure that the MADRS can be accurately used across diverse cultural and linguistic contexts, it must undergo a standardized adaptation process, including two-way translation, testing, revision, and validation [10–13]. This process ensures that the translated version retains the integrity of the original tool while being culturally appropriate. Existing research demonstrates the equivalence of MADRS across several translations [14–19], and highlights its strong psychometric properties in diverse populations [20–25]. However, no studies have yet validated a Vietnamese version of the MADRS, nor has its diagnostic value or psychometric performance been evaluated in RA patients. To address this gap, we conduct this study with the primary aim of validating the Vietnamese version of the MARDS in RA patients.

## Methods

### Study design

A cross-sectional study was conducted from December 2023 to September 2024 at the Center of Rheumatology and Outpatient Department, Bach Mai Hospital in RA patients.

### Patient eligibility

Eligible participants were individuals diagnosed with rheumatoid arthritis according to the American College of Rheumatology (ACR) 1987 criteria. These criteria require the presence of at least four of the following: morning stiffness lasting at least one hour, arthritis in three or more joint areas, arthritis of the hand joints, symmetric arthritis, rheumatoid nodules, serum rheumatoid factor positivity, and radiographic changes typical of RA [26]. Patients with cognitive impairments or disabilities affecting hearing and speech that could hinder effective communication were excluded from the study.

### Data collection

Eligible participants were informed about the study's objectives, procedures, and the data to be collected. After obtaining written consent, interviews and clinical examinations were performed to gather demographic information, including age, gender, average income, occupation, marital status, place of residence, education level, height, and weight. Additionally, RA-related data were collected, including disease duration, the number of hospital admissions due to RA and number of RA flares in the past year, RA medications and their adverse effects, duration of morning stiffness, deformity status, number of swollen joints, and number of tender joints. Disease activity was assessed using the Clinical Disease Activity Index (CDAI), a combined score of number of swollen and tender joints, patient and physician global assessments, which ranged

from 0 to 76. Based on the total CDAI score, disease activity was classified into four categories: Remission: CDAI ≤ 2.8; Low disease activity: CDAI > 2.8 to ≤ 10; Moderate disease activity: CDAI > 10 to ≤ 22; High disease activity: CDAI > 22 [27].

Depression was measured using the Patient Health Questionnaire-9 (PHQ-9) and MADRS. The PHQ-9 consists of nine items, each corresponding to a diagnostic criterion for major depressive disorder as outlined in the DSM-5. Participants rated the frequency of their symptoms over the past two weeks on a scale ranging from 0 (not at all) to 3 (nearly every day). The total score ranged from 0 to 27. Based on the overall PHQ-9 score, participants were classified into one of four groups: no or minimal depressive symptoms (0–9), mild depressive symptoms (10–14), moderate depressive symptoms (15–19), or severe depressive symptoms (20–27) [28].

The MADRS consists of 10 items, each assessing a specific aspect of depression, including apparent sadness, reported sadness, inner tension, reduced sleep, reduced appetite, concentration difficulties, lassitude, inability to feel, pessimistic thoughts, and suicidal thoughts. Each item is scored on a scale from 0 to 6, where 0 indicates no symptoms and 6 reflects the most severe symptom presentation. The total score ranges from 0 to 60, with higher scores indicating more severe depression. The total score is interpreted using the following cut-off points: 0–6: no or very mild depression, 7–19: mild depression, 20–34: moderate depression, and 35–60 points: severe depression [14].

### MARDS translation

The two-way translation process was implemented in this study. The forward translation was conducted by a Vietnamese psychiatrist (T.T.T.H), followed by a back translation back to English by two certified English language experts who were native Vietnamese speakers (N.T.L and L.T.T.H). This back translation was then reviewed by a fourth Vietnamese translator (H.B.L), who possessed a background in both clinical psychiatry and English language. This individual compared the back-translated version with the original instrument, assessing the formatting, terminology, grammatical structure, semantic equivalence, and overall appropriateness. This comparison involved discussions among the research team members, to address any ambiguities and discrepancies that arose during the translation process. A finalized Vietnamese version of the MADRS instrument was developed and subsequently pilot tested on 10 Vietnamese RA patients to assess the clarity and comprehensibility of the translation.

### Sample size and sampling

For this validation study, we adhered to several widely accepted criteria to determine a minimal sample size that met psychometric evaluation requirements. Previous research suggested that a sample size exceeding 100 participants was optimal for evaluating internal consistency, assessing measurement error and reliability, testing construct validity, and conducting subgroup comparisons [29]. A commonly cited rule of thumb was to include at least 10 participants per item in the instrument being tested [10], meaning that for the 10-item MADRS, a minimum of 100 participants was necessary.

Using convenience sampling, all patients with a confirmed diagnosis of RA, without any exclusion criteria, and who provided informed consent, were included in this study.

### Statistical analysis

Continuous variables were presented as mean values (standard deviations), while categorical variables were presented as values and percentages. Stacked bar charts were used to visualize the distribution of responses and depression levels according to both the PHQ-9 and MADRS classifications.

The only type of reliability assessed in this analysis was internal consistency, evaluated using Cronbach's alpha. Additionally, we calculated item-item correlations and item-total correlations using Spearman's r and visualized them using scatter plots.

Face validity was done after the Vietnamese translation had been back-translated and language equivalence had been confirmed.

PLOS Mental Health

Construct validity was first examined using exploratory factor analysis (EFA). The Kaiser-Meyer-Olkin (KMO) measure and Bartlett's test were employed to evaluate sampling adequacy to ensure suitability for EFA. We determined the number of factors using the number of factors before and including the elbow in the Scree plot and parallel analysis. Factor loadings were then calculated using EFA with oblique rotation. Confirmatory factor analysis (CFA) was then utilized to confirm the factor model. Model fit was assessed using the Tucker-Lewis Index (TLI), Comparative Fit Index (CFI), Root Mean Square Error of Approximation (RMSEA), and Standardized Root Mean Square Residual (SRMR). We used the conventional cut-offs for CFA goodness-of-fit (TLI > 0.95, CFI > 0.96, RMSEA <0.08, and SRMR <0.08) that had been described elsewhere. The total Montgomery score was plotted against the total PHQ-9 score and the predicted factor score to demonstrate convergent validity using scatter plots with locally weighted scatterplot smoothing (LOESS) to check for linearity.

A p-value of 0.05 was considered statistically significant. R language version 4.3.2 was used for all analyses. Cronbach's alpha calculation and EFA were done using the `psych` package, and CFA was done using the `lavaan` package.

### Ethical considerations

The study adhered to the guidelines of the Declaration of Helsinki and received approval from the Institutional Review Board of Hanoi Medical University (Decision No. 965/GCN-HĐĐĐNCYSSH-ĐHYHN, dated November 29, 2023). Prior to participation, all patients provided written informed consent, with consent for participants under 18 obtained from their parent or guardian. The investigators maintained compliance with Vietnam's regulations and Good Clinical Practice standards to ensure patient privacy and confidentiality.

## Results

### Participant's characteristics

A total of 120 participants were included in this study, with a mean age of 58.2 ± 11.7 years. The majority were female (83.3%), and most were inpatients (66.7%). A large proportion resided in rural areas (71.7%), and around half had an educational level of high school or below (50.8%). History of physical health conditions were reported by 52.5%, with a smaller percentage having a history of substance use (12.5%) or mental conditions (0.8%). Most participants had experienced RA for over a year (98.3%), and 58.3% had been hospitalized due to RA in the past year. Joint deformity was observed in 40.8%, and around half had high disease activity (49.2%). Common medications used included corticosteroids (64.2%), NSAIDs (46.7%), and DMARDs (48.3%). Adverse drug reactions were reported by 20.8% of participants, with adrenal insufficiency (9.2%) and gastritis/peptic ulcers (5%) being the most common (Table 1).

### Depression measured by the MADRS and PHQ-9

Both the MADRS and PHQ-9 showed similar patterns in depressive symptom severity and suicidal ideation. The prevalence of none, mild, moderate and severe according to MADRS were 47%, 36%, 16% and 1.7%, respectively, while in PHQ-9, they were 80%, 13%, 7.5% and 0%, respectively. Suicidal thoughts were reported by 9% on MADRS and 13% on PHQ-9 (Fig 1, S1 and S2 Figs).

### Reliability

The Cronbach's alpha was 0.93 (95% CI 0.90, 0.94).

The item-item correlations are presented in a heatmap where darker colors represent stronger correlations (S3 Fig). The first two items (Apparent sadness and Reported sadness) and the last two items (Pessimistic thoughts and Suicidal thoughts) were strongly correlated, with Spearman's correlation coefficients r > 0.70. Other items were moderately correlated with one another (r between 0.50 and 0.65).

**Table 1. Participant's charateristics (n = 120).**

| Characteristic | Results |
|---|---|
| Age (years), X ± SD | 58.2 ± 11.7 |
| Gender female, n (%) | 100 (83.3%) |
| Inpatient, n (%) | 80 (66.7%) |
| Location of residence, n (%) | |
| Urban | 33 (27.5%) |
| Rural | 86 (71.7%) |
| Mountainous region | 1 (0.8%) |
| Educational level, n (%) | |
| High school and below | 61 (50.8%) |
| Collegue and above | 59 (49.2%) |
| Marrital status, n (%) | |
| Single/Divorced/Widow | 8 (6.7%) |
| Married | 112 (93.3%) |
| Occupational, n (%) | |
| Worker | 9 (7.5%) |
| Farmer | 37 (30.8%) |
| Trader | 6 (5.0%) |
| Government employee | 6 (5.0%) |
| Housewife | 12 (10.0%) |
| Retired | 26 (21.7%) |
| Student | 1 (0.8%) |
| Self-employed | 17 (14.2%) |
| Unemployed | 3 (2.5%) |
| Others | 3 (2.5%) |
| Individual income ≤ 4.5 millions VND per month, n (%) | 59 (57.3%) |
| Had medical insurance, n (%) | 88 (73.3%) |
| BMI group, n (%) | |
| Underweight | 18 (15.0%) |
| Normal | 84 (70.0%) |
| Overweight | 18 (15.0%) |
| Medical history, n (%) | |
| Physical conditions | 63 (52.5%) |
| Mental conditions | 1 (0.8%) |
| Substance use/abuse | 15 (12.5%) |
| Disease duration ≥ 1 year, n (%) | 118 (98.3%) |
| Hospitalization due to RA in the past year, n (%) | 70 (58.3%) |
| Had RA flare in the past year, n (%) | 50 (41.7%) |
| Morning stiffness duration (minutes), Mean ± SD | 31.9 ± 29.9 |
| Joint deformity, n (%) | 49 (40.8%) |
| Disease activity | |
| Remission | 2 (1.7%) |
| Low disease activity | 13 (10.8%) |
| Moderate disease activity | 46 (38.3%) |
| High disease activity | 59 (49.2%) |
| Medication used, n (%) | |
| NSAIDs | 56 (46.7%) |

*(Continued)*

**Table 1.** (Continued)

| Characteristic | Results |
|---|---|
| Corticosteroids | 77 (64.2%) |
| DMARDs | 58 (48.3%) |
| Biological medications | 34 (28.3%) |
| Medication's AEs | 25 (20.8%) |
| Adrenal insufficiency | 11 (9.2%) |
| Gastritis/Peptic ulcer | 6 (5.0%) |
| Cushing's syndrome | 4 (3.3%) |
| Osteoporosis | 4 (3.3%) |
| Peripheral edema | 1 (0.8%) |
| Kidney failure | 1 (0.8%) |
| Elevated liver enzyme | 2 (1.7%) |

Abbreviations: SD, standard deviation; VND, Vietnam dong; RA, rheumatoid arthritis; NSAIDs, nonsteroidal anti-inflammatory drugs; DMARDs, disease-modifying antirheumatic drugs; BMI, body mass index; GI, gastrointestinal.

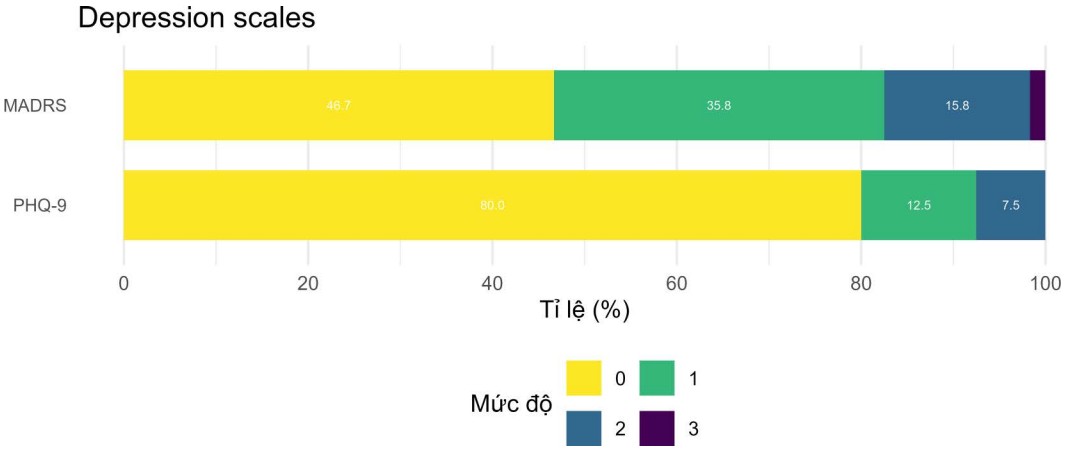

**Fig 1. Depression severity according to PHQ-9 and MADRS (n = 120).**

The item-total correlations are presented in a series of scatter plots (S4 Fig). In general, there was a strongly linear trend between the item and the total score in most items, except the Suicidal thoughts item (probably due to the low numbers of observations with high scores in the item).

## Validity

A team including psychiatrists, language experts, epidemiologists, and non-medical people discussed the Vietnamese translation. The Vietnamese version of the MADRS demonstrated strong equivalence to the original English version in terms of content, structure, and meaning. The translation process, which included forward and back translation, confirmed that all items, instructions, and response options accurately reflected the original tool. A thorough review by a multidisciplinary team ensured that the translated version retained the same conceptual and semantic integrity. The psychiatrist confirmed that the items remained clinically relevant for assessing depressive symptoms in Vietnamese patients with RA,

while language experts validated the accuracy and natural flow of the translation. Non-medical reviewers also confirmed that the instrument was easy to understand. No major discrepancies were identified, and minor adjustments were made to ensure that cultural nuances were appropriately addressed. Overall, the translated MADRS was deemed equivalent to the original in terms of content, clarity, and cultural appropriateness.

The KMO index was 0.92 and the Bartlett's test p-value was < 0.001, suggesting that the dataset was adequate for EFA. Both the Scree plot and parallel analysis (S5 Fig) showed that a model with two factors was appropriate. In the EFA model with two factors, items 1–8 and items 9–10 had high loadings in factor 1 and factor 2, respectively (Table 2 and S6 Fig). Based on the nature of the items, we named these two factors "Depressive symptoms" and "Pessimism".

After running the EFA, we fitted an initial CFA model with the same structure as the EFA model. Goodness-of-fit indices demonstrated that the model did not achieve an adequate fit (CFI 0.956; TLI 0.942; RMSEA 0.093, 90% CI 0.0, 0.125, $p < 0.001$; SRMR 0.044). We used the modification indicies to improve the model fit. A new model with covariance between the first two items (Apparent sadness and Reported sadness) demonstrated an adequate fit (CFI 0.961; TLI 0.947; RMSEA 0.090, 90% CI 0.057, 0.122, $p = 0.026$; SRMR 0.043).

We then plotted the total MARDS score against the total PHQ-9 score and the two predicted factor scores calculated from the final CFA model. The total MARDS score was strongly correlated with the total PHQ-9 score (Spearman's r 0.88); this correlation could be considered linear based on the LOESS plot (Fig 2). The total MARDS score was strongly and linearly correlated with the Depression symptoms factor; however, it was not correlated well with the Pessimism factor (Fig 3). Because of the strong correlation between the total score and the Depression symptoms factor score, we examined the total MARDS score excluding the last two items (which belonged to the Pessimism factor). This "excluded" total score were strongly correlated with the total MARDS score (S7 Fig).

## Discussion

The primary aim of this study was to validate the Vietnamese version of the MADRS instrument the RA population. In our study, the Vietnamese-translated MADRS demonstrated excellent internal consistency, with a Cronbach's alpha of 0.93. Similar strong alpha values were reported in other translated versions of the MADRS, included the Colombian (α=0.9168) [17], Persian (α=0.9) [18], Bangla (α=0.87) [19], Spanish (α=0.88) [30], and German (α=0.9) [22] versions. Although the Brazilian (α=0.7–0.84) [15] and Malaysian (α=0.78) [16] versions showed slightly lower values, these still fall within the acceptable range for internal consistency (α>0.7) [31]. The lower alpha values in some translations may be influenced by subtle cultural or linguistic differences affecting how patients interpret specific items, leading to minor variations in responses [32–34].

**Table 2. Factor loadings for MARDS items.**

| Item | Factor 1 | Factor 2 |
|---|---|---|
| Item 1: Apparent Sadness | **0.50** | 0.33 |
| Item 2: Reported Sadness | **0.75** | 0.14 |
| Item 3: Inner Tension | **0.81** | 0.07 |
| Item 4: Sleep | **0.65** | 0.07 |
| Item 5: Appetite | **0.93** | -0.17 |
| Item 6: Concentration | **0.79** | 0.05 |
| Item 7: Lassitude | **0.48** | 0.23 |
| Item 8: Feel | **0.82** | 0.00 |
| Item 9: Pessimistic | 0.14 | **0.78** |
| Item 10: Suicidal | -0.04 | **0.85** |

Bold values are loadings >0.4.

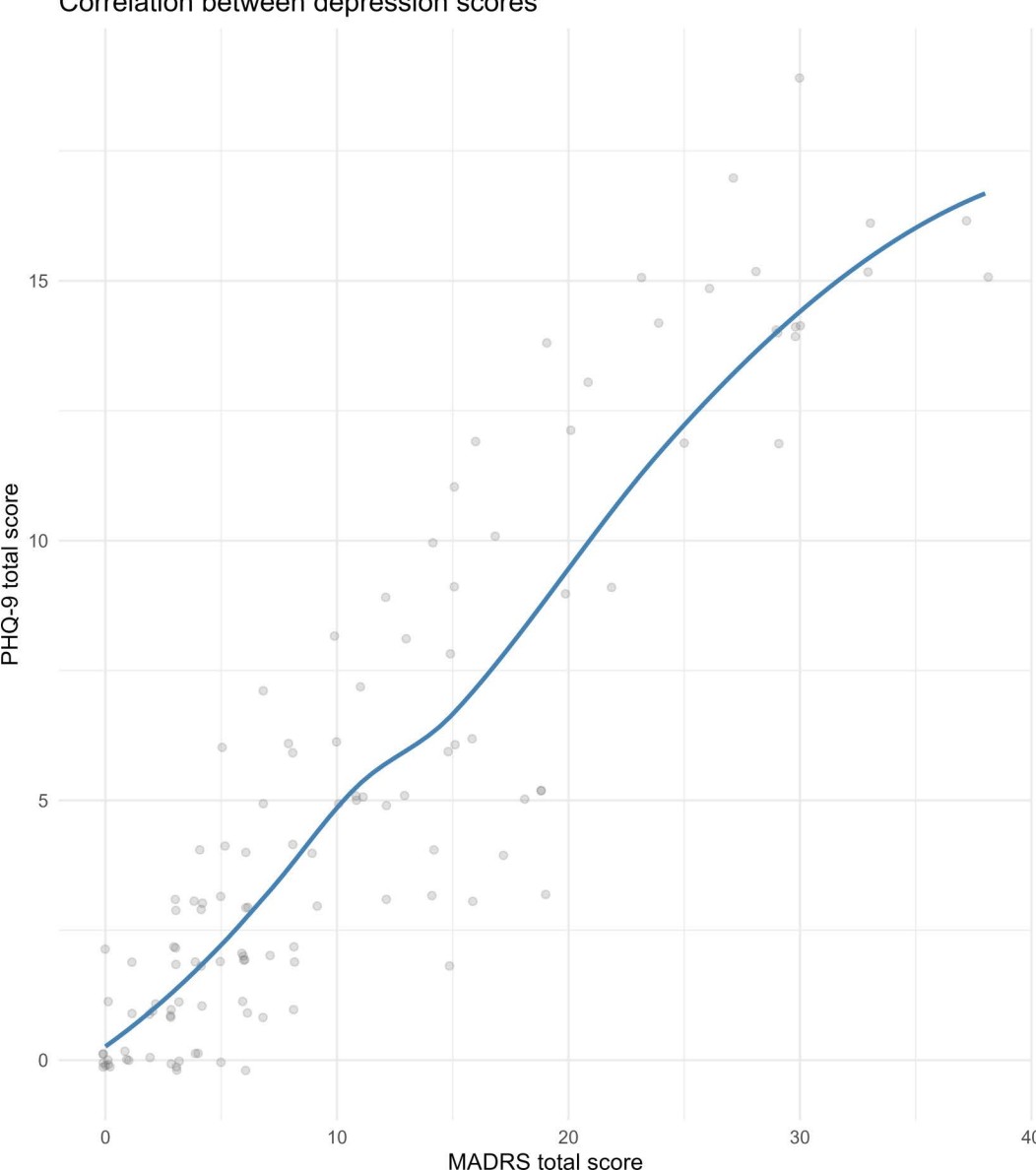

**Fig 2. Correlation between MADRS and PHQ-9.**

However, these differences do not necessarily indicate poor reliability, as the Malaysian study showed that removing items did not improve the alpha value, confirming that the MADRS retains its internal consistency in this translated version [16].

Our study also demonstrated the convergent validity of the Vietnamese version of the MADRS, as it was strongly and linearly correlated with the PHQ-9. This correlation between the MADRS and PHQ-9 was consistently reported in previous studies [35–37], likely because both instruments measure overlapping core symptoms of depression, namely sadness, loss of interest, and suicidal ideation [9,28]. Additionally, literature documented correlations between the MADRS and other widely used depression instruments, including the Hamilton Depression Rating Scale (HAMD) [18,22,38–40], Beck Depression Inventory (BDI) [22,24,41,42], and the Hospital Anxiety and Depression Scale (HADS) [24,43]. These scales,

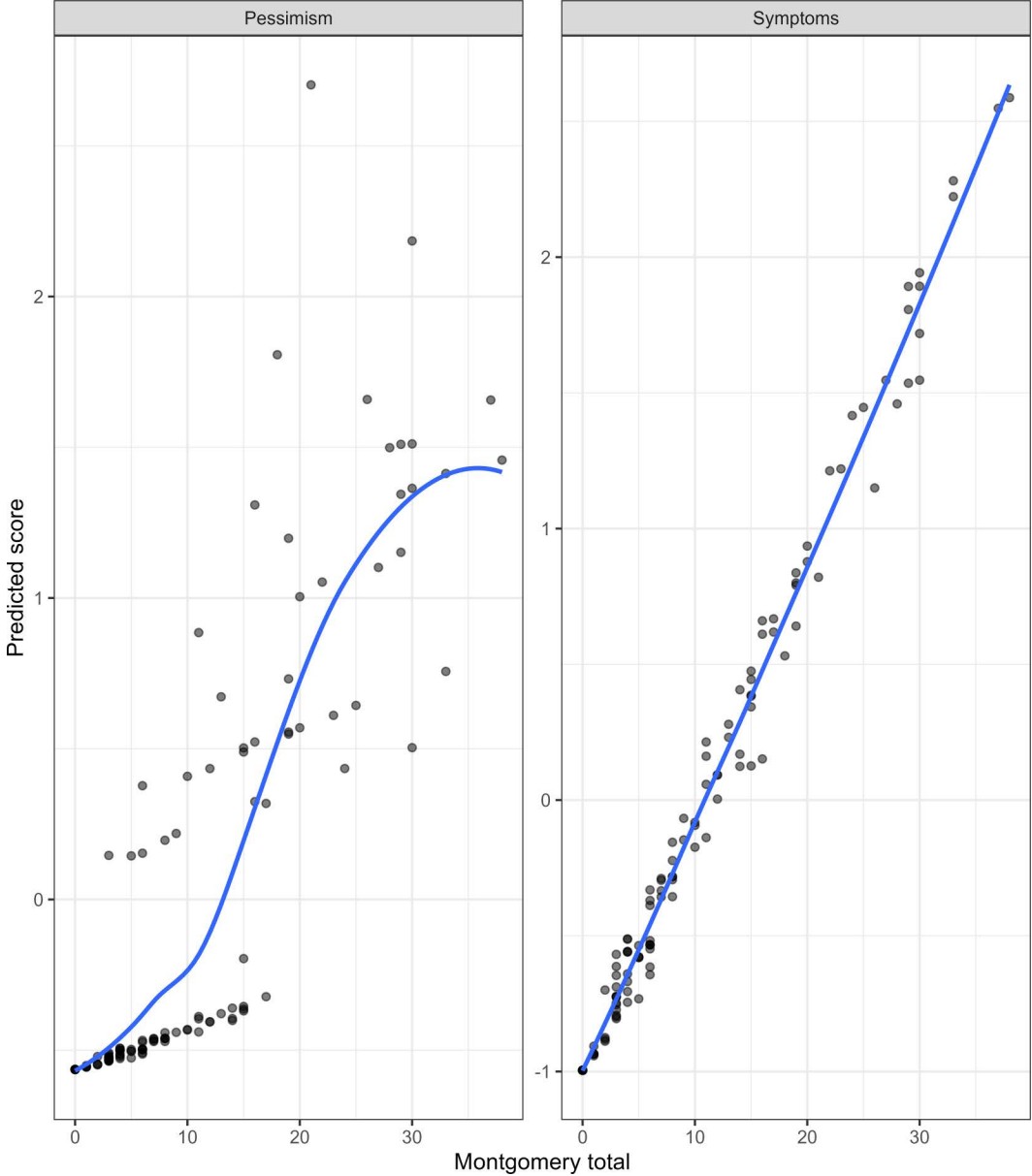

**Fig 3.** **Correlation between MARDS total score and predicted factor scores.**

similar to the MADRS, assess multiple dimensions of depression, including emotional, cognitive, and somatic symptoms, which accounts for their strong correlations. The consistent associations across different instruments highlight that while the specific items and scoring systems may vary, these tools ultimately capture the same underlying construct of depressive symptomatology, reinforcing the validity of the MADRS in diverse clinical settings.

In the factor analysis, a two-factor structure was identified within the Vietnamese MADRS. First factor represented general depressive symptoms, while the second factor associated with more severe affective states, including "Suicidal" and "Pessimistic". Previous studies has demonstrated a variety of factor structures within the MADRS across different populations and settings. An unidimensional structure has often been reported, which involved general population [16], patients

with coronary artery disease [44], and depression [45]. On the other hand, previous studies have identified a two-factor structure, however, the composition of factors varied depending on the characteristics of the population. In a 2010 study involving 221 patients diagnosed with early-onset dementia, factor analysis revealed a two-factor structure. The first factor, dysphoria, encompassed symptoms such as impaired concentration, pessimistic thoughts, inner tension, suicidal ideation, lassitude, and sleep disturbances. The second factor, sadness, included observed sadness, reported sadness, reduced appetite, and inability to feel [46]. In another study which included 100 elderly with medical comorbidities, factor analysis revealed two factor: "Anhedonia" (including Lassitude, Inability to feel, Apparent sadness, Concentration loss) and "Dysphoria" (including inner tension, Reported sadness, Suicidal feelings) [47]. Three- and four-factor models have also been reported across various populations with differing conditions [48–52].

The identification of multiple factors is consistent with the multifaceted nature of depression, as symptoms can manifest differently based on individual and contextual differences [53,54]. The differences in the number and nature of factor structures observed across studies may be attributed to variations in the demographic and clinical characteristics of the populations studied. Further, methods for determining the number of factors and the conditions under which to rotate extracted factors have varied. The two-factor structure observed in this study supports the notion that depressive symptoms can be categorized into general and more severe clusters, particularly in the context of RA. Clinically, this distinction may aid in developing more targeted interventions, where specific symptoms identified within the severe cluster, could be directly addressed to reduce the risk of progressing to more severe depressive states. Additionally, recognizing a two-factor structure may provide a clearer understanding of symptom progression, as patients may move from general to more severe symptoms if their depression worsens. However, the "severe" factor may not be adequately represented in the current structure of the questionnaire and will thus not be measured accurately.

There were several limitations to consider in this study. First, our population differed significantly from the other RA populations, as it primarily consisted of inpatients with prolonged disease duration and medication use, a high incidence of disease and medication-related complications, and a high disease activity levels. This composition posed a threat to the generalizability of the findings. Second, the study did not employ a gold standard diagnostic tool (such as ICD-10 criteria for depression) to establish convergent validity, which limited our ability to estimate diagnostic value and determine clinically relevant MADRS cut-off points. Lastly, due to resource constraints, we were unable to conduct a longitudinal follow-up to assess test-retest reliability, nor could we involve multiple independent raters to evaluate inter-rater reliability. These are essential psychometric properties for evaluating the reliability of this instrument in clinical practice.

## Conclusion

This study demonstrated that the Vietnamese version of the MADRS was a reliable and valid tool for assessing depressive symptoms in RA population. Although limitations such as the specific inpatient sample and lack of a gold standard diagnostic tool affected generalizability, the strong internal consistency and promising validity suggested that the Vietnamese MADRS could be effectively utilized in both clinical and research settings in Vietnam. Future studies should focus on expanding the sample diversity and exploring additional reliability measures to further validate the tool.

## Supporting information

**S1 Fig. Distribution of the MADRS responses among participants (n = 120).**
(TIF)

**S2 Fig. Distribution of the PHQ-9 responses among participants (n = 120).**
(TIF)

**S3 Fig. Item-item correlation of the MARDS.**
(TIF)

**S4 Fig. Item-total correlation of the MADRS.**
(TIF)

**S5 Fig. Scree plot and Parallel analysis.**
(TIF)

**S6 Fig. Loadings plot for the EFA model with two factors.**
(TIF)

**S7 Fig. Correlation between the total MADRS score and the modified total score excluding pessimism factor items.**
(TIF)

## Author contributions

**Conceptualization:** Ha Thi Thu Tran, Long Bao Hoang, Hung Van Nguyen, Tuan Van Nguyen, Ha Thi Thu Le, Yen Hoang Nguyen, Hoa Thi Nguyen, Long Thanh Nguyen, Tam Minh Duong.

**Data curation:** Ha Thi Thu Tran.

**Formal analysis:** Ha Thi Thu Tran, Long Bao Hoang, Tam Minh Duong.

**Funding acquisition:** Ha Thi Thu Tran.

**Investigation:** Ha Thi Thu Tran, Hoa Thi Nguyen, Hue Thi Doan, Khiem Tuan Ngo, Thien Cong Le, Duong Thuy Lai, Tung Son Vu, Nga Thi Nguyet Pham, Tam Minh Duong.

**Methodology:** Ha Thi Thu Tran, Long Bao Hoang, Yen Hoang Nguyen, Hoa Thi Nguyen, Hue Thi Doan, Khiem Tuan Ngo, Thien Cong Le, Cam Thy Vu, Duong Thuy Lai, Tung Son Vu, Long Thanh Nguyen, Nga Thi Nguyet Pham, Tam Minh Duong.

**Project administration:** Ha Thi Thu Tran, Hung Van Nguyen, Tuan Van Nguyen, Cam Thy Vu, Duong Thuy Lai, Long Thanh Nguyen, Tam Minh Duong.

**Resources:** Ha Thi Thu Tran.

**Supervision:** Ha Thi Thu Tran, Hung Van Nguyen, Tuan Van Nguyen, Ha Thi Thu Le.

**Validation:** Ha Thi Thu Tran.

**Visualization:** Ha Thi Thu Tran, Yen Hoang Nguyen.

**Writing – original draft:** Ha Thi Thu Tran, Long Bao Hoang, Tam Minh Duong.

**Writing – review & editing:** Ha Thi Thu Tran, Long Bao Hoang, Hung Van Nguyen, Tuan Van Nguyen, Ha Thi Thu Le, Yen Hoang Nguyen, Hoa Thi Nguyen, Hue Thi Doan, Khiem Tuan Ngo, Thien Cong Le, Cam Thy Vu, Duong Thuy Lai, Tung Son Vu, Long Thanh Nguyen, Nga Thi Nguyet Pham, Tam Minh Duong.

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
