## [Decision Letter · Decision Letter 0]

PMEN-D-25-00046

Validation of the Vietnamese Version of the Montgomery-Asberg Depression Rating Scale (MADRS) in Rheumatoid Arthritis Patients

PLOS Mental Health

Dear Dr. Tran,

Thank you for submitting your manuscript to PLOS Mental Health. I am sorry for the delay. After careful consideration of the reviewer reports, we feel that your paper has merit but does not yet fully meet PLOS Mental Health’s publication criteria as it currently stands. Therefore, we invite you to submit a revised version of the manuscript that addresses the points raised during the review process.

Please ensure that you address all of the comments raised by the reviewers, which you can find at the end of this email.

We look forward to receiving your revised manuscript.

Kind regards,

Karli Montague-Cardoso

Executive Editor

PLOS Mental Health

Journal Requirements:

1. We noticed you have some minor occurrence of overlapping text with the following previous publication(s), which needs to be addressed:

- https://doi.org/10.3389/fpsyt.2019.00321

- doi: 10.1186/1477-7525-6-2

- doi: 10.1186/1741-7015-10-156

In your revision ensure you cite all your sources (including your own works), and quote or rephrase any duplicated text outside the methods section. Further consideration is dependent on these concerns being addressed.

2. Please provide separate figure files in .tif or .eps format.

https://journals.plos.org/mentalhealth/s/figures

https://journals.plos.org/mentalhealth/s/figures#loc-file-requirements

3. In the online submission form, you indicated that [The datasets generated during and/or analysed during the current study are available from the corresponding author on reasonable request.].

a. In a public repository,

b. Within the manuscript itself, or

c. Uploaded as supplementary information.

Additional Editor Comments (if provided):

Reviewers' comments:

Reviewer's Responses to Questions

**Comments to the Author**

1. Does this manuscript meet PLOS Mental Health’s publication criteria ? Is the manuscript technically sound, and do the data support the conclusions? The manuscript must describe methodologically and ethically rigorous research with conclusions that are appropriately drawn based on the data presented.

Reviewer #1: Yes

Reviewer #2: Yes

2. Has the statistical analysis been performed appropriately and rigorously?

Reviewer #1: Yes

Reviewer #2: Yes

3. Have the authors made all data underlying the findings in their manuscript fully available (please refer to the Data Availability Statement at the start of the manuscript PDF file)?

Reviewer #1: Yes

Reviewer #2: Yes

4. Is the manuscript presented in an intelligible fashion and written in standard English?

Reviewer #1: Yes

Reviewer #2: Yes

5. Review Comments to the Author

Reviewer #1: Dear Authors,

I would like to express my gratitude for your valuable research contributions.

1. Could you elucidate the rationale behind your selection of this specific tool? Is it intended for the diagnosis or screening of depression, and does it possess validity in assessing depression among patients with rheumatoid arthritis (RA)?

2. In the introduction section, it would be beneficial to provide a concise overview of the pathophysiological relationship between RA and depression. Is this relationship primarily mediated by cytokine activity or chronic pain mechanisms?

3. I am curious as to why you predominantly relied on an older set of criteria for RA classification. The presence of these criteria suggests that the patient is experiencing advanced RA, which may subsequently influence their psychological well-being.

4. Considering that RA patients are susceptible to various psychological issues such as fatigue, anxiety, and fibromyalgia, how did you ascertain that these disorders do not affect the disease activity of RA?

5. Why was a laboratory measurement, such as erythrocyte sedimentation rate (ESR) or C-reactive protein (CRP), not included to monitor disease activity in RA?

6. Your findings indicate that 38% of your sample exhibits moderate disease activity and 50% demonstrates high disease activity; however, less than 17% of your patients reported moderate to severe depression. This discrepancy raises questions.

7. While you acknowledged inter-rater reliability as a limitation in assessing the reliability of your instrument, you ultimately concluded that it is a reliable and valid tool for evaluating depression. Could you clarify this apparent contradiction?

Reviewer #2: The manuscript entitled "Validation of the Vietnamese Version of the Montgomery-Asberg Depression

Rating Scale (MADRS) in Rheumatoid Arthritis Patients" done by Ha Thi Thu Tran et al. is interesting in the field of depression evaluation. In this study, the authors aimed to validate the Vietnamese version of the Montgomery Asberg Depression Rating Scale (MADRS) in patients with rheumatoid arthritis. They did a cross-sectional study on RA patients, with depression severity assessed using both MADRS and Patient Health Questionnaire-9 (PHQ-9). They found that Vietnamese MADRS demonstrated excellent internal consistency and had a significant linear correlation with PHQ-9. They also identified a two-factor structure, categorizing depressive symptoms into general and severe clusters. The authors concluded that the Vietnamese MADRS can be effectively used in clinical and research settings in Vietnam for assessing depression in RA patients.

However, there are some issues which should be clarified by the authors:

1- MADRS translation:

- The final Vietnamese version of two-way translated MARDS should be validated by the scientific committee in the field such as local society or association or department of psychology for reusing in practice.

- The authors should have a permission to use and translate MADRS from the the owners.

- The authors should define who did pilot tested on 10 Vietnamese RA patients to assess the clarity and comprehensibility of the translation.

- The Vietnamese version of MADRS should be done in appendix.

2- Results:

- Table 1 is too long. It could be presented and multiple semi-colon with the same way

3- Discussion:

- The authors should discuss why other scales using to evaluated depression such as Hamilton Rating Scale for Depression, or Pichot, etc...did not use foe this study population?

- The authors should discuss deeply about why the prefer to use MADRS in RA because in its history, MADRS was designed in 1979 by Stuart Montgomery and Marie Åsberg as an adjunct to the Hamilton Rating Scale for Depression (HAMD) and MADRS has been suggested being more sensitive to the changes brought on by antidepressants and other forms of treatment than the Hamilton Scale.

6. PLOS authors have the option to publish the peer review history of their article (what does this mean? ). If published, this will include your full peer review and any attached files.

**Do you want your identity to be public for this peer review?** For information about this choice, including consent withdrawal, please see our Privacy Policy .

Reviewer #1: No

Reviewer #2: **Yes: ** Sy Duong-Quy

---

## [Editor Report · Decision Letter 1]

Validation of the Vietnamese Version of the Montgomery-Asberg Depression Rating Scale (MADRS) in Rheumatoid Arthritis Patients

PMEN-D-25-00046R1

Dear MD Tran,

We are pleased to inform you that your manuscript 'Validation of the Vietnamese Version of the Montgomery-Asberg Depression Rating Scale (MADRS) in Rheumatoid Arthritis Patients' has been provisionally accepted for publication in PLOS Mental Health.

Best regards,

Karli Montague-Cardoso

Staff Editor

PLOS Mental Health